# Active learning of reactive Bayesian force fields applied to heterogeneous catalysis dynamics of H/Pt

Jonathan Vandermause [1,2] ✉, Yu Xie [2], Jin Soo Lim [3], Cameron J. Owen [3] & Boris Kozinsky [2,4] ✉

Atomistic modeling of chemically reactive systems has so far relied on either expensive ab initio methods or bond-order force fields requiring arduous parametrization. Here, we describe a Bayesian active learning framework for autonomous "on-the-fly" training of fast and accurate reactive many-body force fields during molecular dynamics simulations. At each time-step, predictive uncertainties of a sparse Gaussian process are evaluated to automatically determine whether additional ab initio training data are needed. We introduce a general method for mapping trained kernel models onto equivalent polynomial models whose prediction cost is much lower and independent of the training set size. As a demonstration, we perform direct two-phase simulations of heterogeneous $H_2$ turnover on the Pt(111) catalyst surface at chemical accuracy. The model trains itself in three days and performs at twice the speed of a ReaxFF model, while maintaining much higher fidelity to DFT and excellent agreement with experiment.

Accurate modeling of chemical reactions is a central challenge in computational physics, chemistry, and biology, lying at the heart of in silico design of covalent drugs[1] and next-generation catalysts with higher activity and selectivity[2]. Reactive molecular dynamics (MD) simulation is an essential tool in advancing such rational design efforts[3]. By directly simulating the motion of individual atoms without fixing any chemical bonds, reactive MD enables unbiased discovery of reaction mechanisms at atomic resolution as well as prediction of reaction rates complementary to experimental studies[4].

Reactive MD requires a flexible model of the potential energy surface (PES) of the system that is both (i) chemically accurate in describing bond breaking and formation, and (ii) computationally affordable to be able to access long timescales necessary to capture rare reactive events. Accurate evaluation of the PES at each time-step can be achieved with ab initio methods such as density functional theory (DFT) and post-Hartree-Fock techniques. However, these methods are limited to small systems due to nonlinear scaling with the number of electrons, precluding their use for dynamical simulation of realistic systems beyond a few hundred atoms spanning tens of picoseconds.

For many nonreactive systems, a viable alternative to ab initio MD is to parameterize a force field whose evaluation cost scales linearly with the number of atoms and is often orders of magnitude cheaper than ab initio methods. However, many traditional force fields, such as the AMBER and CHARMM models extensively used in biomolecular simulations[5,6], explicitly fix certain chemical bonds in the system, making them unsuitable for describing chemical reactions. The more flexible ReaxFF model is capable of describing bond breaking and formation and as such has been applied to a wide range of reactive systems in the past two decades[3,7]. However, ReaxFF models require expert fine-tuning for each system and can significantly deviate from the ab initio PES in many cases[8] due to their limited parametric functional form.

Machine-learned (ML) force fields have emerged in the past decade as a powerful tool for building linear-scaling models of the PES at near-DFT accuracy[9–12]. These ML models map atomic configurations

[1]Department of Physics, Harvard University, Cambridge, MA 02138, USA. [2]John A. Paulson School of Engineering and Applied Sciences, Harvard University, Cambridge, MA 02138, USA. [3]Department of Chemistry and Chemical Biology, Harvard University, Cambridge, MA 02138, USA. [4]Robert Bosch LLC, Research and Technology Center, Cambridge, MA 02139, USA. ✉e-mail: jonathan_vandermause@g.harvard.edu; bkoz@seas.harvard.edu

onto potential energies, atomic forces, and virial stress tensors, using ab initio calculations as the ground truth for the regression. Typically, training these models involves manual construction of the ab initio database targeting the system of interest. This manual approach has been used to describe a range of simple bulk materials[8,13,14] and organic molecules[15–17]. However, manual training often requires considerable time, expertise, and computing resources, and is particularly challenging for reactive systems where relevant transition state pathways and their sampling requirement can be difficult to gauge in advance. As a result, ML-driven MD simulations of reactive processes remain scarce in the literature[18–22].

A powerful emerging alternative is to generate the training set autonomously using active learning[23–28]. In this approach, model errors or uncertainties are used to decide if a candidate test structure is reliably predicted or should be added to the training set, in which case an ab initio calculation is performed and the model is updated. A particularly promising approach involves training the model "on the fly" during an MD simulation, where the ML model is used to propagate atomic motion and is updated only when the model uncertainties exceed a chosen threshold[29–32]. Active learning has been applied successfully in the past year to a range of nonreactive systems and phenomena, including phase transitions in hybrid perovskites[33], melting points of solids[34], superionic transport in silver iodide[32], surface restructuring of palladium deposited on silver[35], the 2D-to-3D phase transition of stanene[36], and lithium-ion diffusion in solid electrolytes[37]. It has also been extended recently to chemical reactions within the Gaussian approximation potential (GAP) framework[21]. The kernel-based GAP force field, although highly accurate, has a prediction cost that scales linearly with the size of the sparse set, making it several orders of magnitude more expensive than traditional force fields such as ReaxFF[38].

Chemically reactive systems pose a particular challenge for machine-learned force fields. Owing to its central importance in numerous catalytic processes such as selective hydrogenation[39] and hydrogen storage[40], $H_2$ reactivity on transition metal surfaces has been a subject of extensive computational investigation, including DFT calculations of $H_2$ activation and diffusion[41–43], as well as MD simulations using DFT[44] and parametric models such as ReaxFF[45], embedded-atom method[46], tight-binding[47], and low-dimensional models[48]. We note that all previous models only consider either a bare surface interacting with a single $H_2$ molecule or H-covered surface coupled to an implicit reservoir representing the gas phase. These approaches do not explicitly treat the two interacting phases due to either high cost or limited expressiveness insufficient for capturing many-body interactions in a heterogeneous setting. This limits their transferability to realistic systems involving multiple gas-phase $H_2$ molecules and chemisorbed H atoms.

Here, we develop an autonomous method for training reactive many-body force fields on the fly and accelerating the resulting ML models by over an order of magnitude. We apply our method to large-scale reactive MD simulations of a prototypical system in the field of heterogeneous catalysis: reactive turnover of $H_2$ on the (111) surface of platinum, including dissociative adsorption, diffusion, exchange, and recombinative desorption. For the H/Pt system, our prediction speed exceeds that of ReaxFF by more than a factor of two while achieving much higher near-quantum accuracy. Importantly, the training process takes only a few days compared to the months previously required with manual approaches. We accomplish this by using Bayesian uncertainties of a sparse Gaussian process (SGP) model to automatically decide which structures to include in the training set. Once the model is trained, the prediction speed is significantly increased via lossless mapping of the SGP model onto an equivalent parametric model whose cost is independent of the training set size. Our method builds on the sparse Gaussian process force fields introduced in refs. 10,49, and the SGP-based on-the-fly training methods of refs. 33, 34,

extending these methods to a canonical chemically reactive system and establishing their equivalence to a simpler class of polynomial models. Our method also builds on our own previous active learning workflow, which relied on a significantly more expensive exact Gaussian process[32] and was limited to two- and three-body interactions that are insufficiently descriptive for chemically reactive systems. Our active learning approach overcomes these limitations through efficient data-driven construction of the PES for the entire H/Pt system, at high fidelity to the chosen ab initio method. To our best knowledge, we perform the first large-scale direct MD simulations of reactive $H_2$ turnover on Pt(111), capturing the explicit two-phase boundary across the molecular gas phase and fully thermalized substrate with varying H coverages. Importantly, the procedure requires no simplifications or prior assumptions about the reaction mechanisms, proceeding entirely autonomously and providing direct computational measurement of the catalytic reaction kinetics. The resulting apparent activation energy for $H_2$ turnover is found to be in excellent agreement with surface science experiments.

## Results

### Active learning and acceleration of many-body force fields

Figure 1 shows an overview of our method for autonomous training and acceleration of many-body Bayesian force fields. The on-the-fly training loop (Fig. 1a) is driven by an SGP that provides Bayesian uncertainties of model predictions. The objective is to automatically construct both the training set of the SGP and the sparse set, which is a collection of representative local atomic environments $\rho_t$ that are summed over at test time to make predictions and evaluate uncertainties (see Methods). The training loop begins with a DFT calculation on an initial atomic structure, initializing the training set and serving as the first frame of the MD simulation.

At each time-step of the MD simulation, the SGP predicts the potential energy, forces, and stresses of the current structure and assigns Bayesian uncertainties to local energy predictions $\varepsilon$. These uncertainties take the form of a scaled predictive variance $\widetilde{V}_\varepsilon$ valued between 0 and 1 and are defined to be independent of the hyperparameters of the SGP kernel function (see Eq. (17), Methods). This formulation provides a robust measure of the distance between the local atomic environments $\rho_i$ observed in the training simulation and the environments $\rho_t$ stored in the sparse set of the SGP. If the uncertainty is below a chosen "prediction threshold" $\Delta_{DFT}$, the predictions of the SGP are accepted and an MD step is taken using the model forces.

When the uncertainty exceeds the prediction threshold, the simulation is halted and a DFT calculation is performed on the current structure. The computed DFT energy, forces, and stresses are then added to the training set of the SGP, and local environments $\rho_i$ with uncertainty above an "update threshold" $\Delta_{sparse} \le \Delta_{DFT}$ are added to the sparse set. This active selection of sparse points reduces redundancy in the model by ensuring that only sufficiently novel local environments are added to the sparse set. It also helps to reduce the cost of SGP prediction, which scales linearly with the size of the sparse set (see Eq. (9), Methods). The training simulation is terminated when calls to DFT become infrequent, typically after 3–10 ps of dynamics.

The accuracy of the learned force field is in large part determined by the expressiveness of the descriptor vectors assigned to local atomic environments. Our procedure for mapping local environments $\rho_i$ onto symmetry-preserving many-body descriptors $\mathbf{d}_i$ draws on the atomic cluster expansion (ACE) introduced by Drautz[50] and is sketched in Fig. 1b. A rotationally equivariant descriptor $\mathbf{c}_i$ (see Eq. (4), Methods) is first computed by passing interatomic distance vectors through a basis set of radial functions and spherical harmonics and summing over all neighboring atoms of a particular species. Rotationally invariant contractions of the tensor product $\mathbf{c}_i \otimes \mathbf{c}_i$ are then collected in an array $\mathbf{d}_i$ of many-body invariants, which serves as input to the SGP

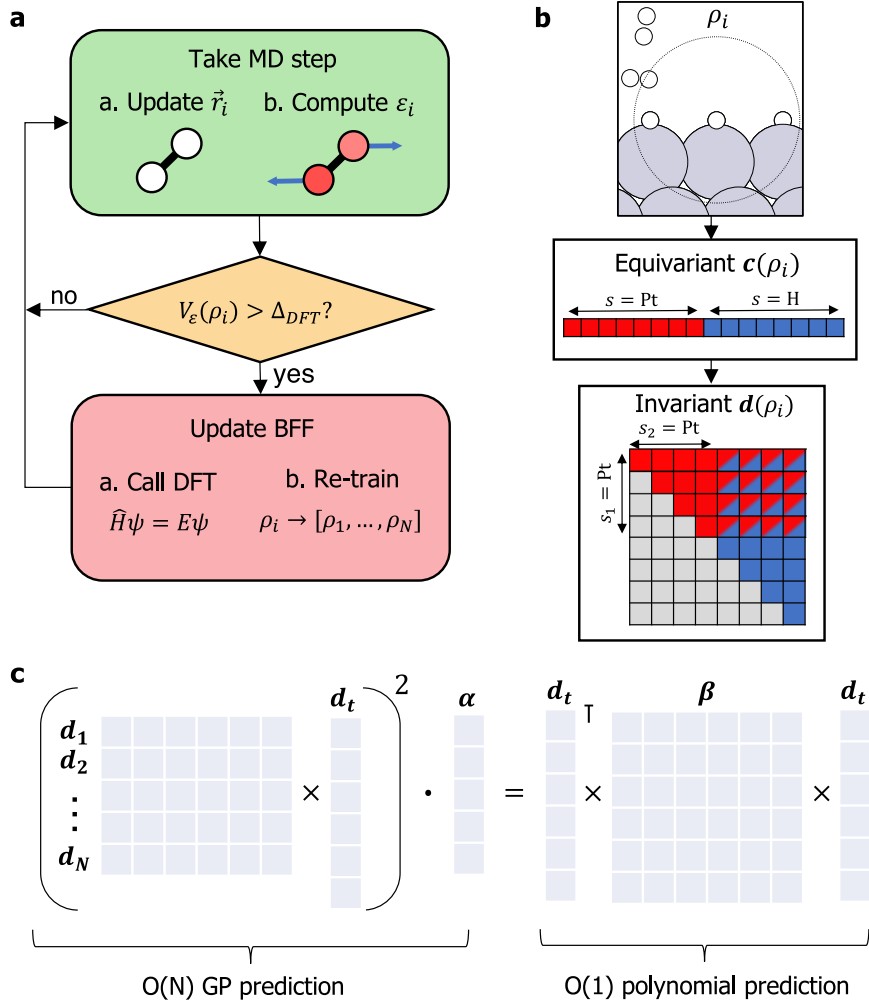

**Fig. 1 | On-the-fly training and acceleration of many-body Bayesian force fields (BFF). a** At each time-step of the MD simulation, local energies, forces, and stresses are computed with the SGP model. If the uncertainty on a local energy exceeds the chosen threshold $\Delta_{DFT}$, a DFT calculation is performed and the model is updated. **b** Mapping of local environments $\rho_i$ onto multielement descriptors derived from the atomic cluster expansion. The environment is first mapped onto an equivariant descriptor $\mathbf{c}_i$, products of which are used to compute the rotationally invariant descriptor $\mathbf{d}_i$ that serves as an input to the model. **c** Mapping of a $\xi = 2$ SGP force field onto an equivalent quadratic model. The prediction cost of the SGP scales linearly with the number of sparse environments $N_S$, while the cost of the corresponding polynomial model is independent of $N_S$.

model. The vector $\mathbf{d}_i$ corresponds to the $B_2$ term in the multielement atomic cluster expansion and is closely related to the SOAP descriptor[51]. In each of these approaches, the number of elements of $\mathbf{d}_i$ scales quadratically with the number of chemical species in the system.

Crucially, once sufficient training data have been collected, we map the trained SGP model onto an equivalent and much faster model whose prediction cost is independent of the size of the sparse set (Fig. 1c). This mapping draws on the duality in machine learning between kernel-based models on the one hand, which make comparisons with the training set at test time and are therefore "memory-based," and linear parametric models on the other, which are polynomials of the features and do not depend explicitly on the training set once the model is trained[52].

As shown in the Methods section, mean predictions of an SGP trained with a normalized dot product kernel raised to an integer power $\xi$ can be evaluated as a polynomial of order $\xi$ in the descriptor $\mathbf{d}_i$. The integer $\xi$ determines the body-order of the learned force field, which we define following ref. 53 to be the smallest integer $n$ at which the derivative of the local energy with respect to the coordinates of any $n$ distinct neighboring atoms vanishes. The simplest case, $\xi = 1$, corresponds to a model that is linear in $\mathbf{d}_i$, and since the elements of $\mathbf{d}_i$ are

sums of three-body contributions, the resulting model is formally three-body. $\xi = 2$ models are quadratic in $\mathbf{d}_i$ and thus five-body, with general $\xi$ corresponding to a $(2\xi + 1)$-body model when using the $B_2$ term of ACE.

We evaluate the performance of kernels with different $\xi$ values by comparing the log marginal likelihood $\mathcal{L}(\mathbf{y}|\xi)$ of SGP models trained on the same structures. $\mathcal{L}$ quantifies the probability of the training labels $\mathbf{y}$ given a particular choice of hyperparameters and can be used to identify hyperparameters that optimally balance model accuracy and complexity (see Eq. (19), Methods). For H/Pt(111), we find that the likelihood $\mathcal{L}(\mathbf{y}|\xi)$ for $\xi = 2$ is considerably higher than for $\xi = 1$ but nearly the same as for $\xi = 3$, and decreases for $\xi > 3$ (Supplementary Fig. 1). We therefore choose to train five-body $\xi = 2$ models, allowing local energies to be evaluated as a simple vector-matrix-vector product $\varepsilon(\rho_i) = \mathbf{d}_i \boldsymbol{\beta} \mathbf{d}_i$ that can be rapidly computed (see Eq. (20) for the definition of $\boldsymbol{\beta}$). We have implemented this quadratic model as a custom pair-style in the LAMMPS code, which exhibits a dramatic acceleration over standard SGP mean prediction (Supplementary Fig. 4). Remarkably, the resulting LAMMPS model is more than twice as fast as a recent H/Pt ReaxFF model[45], opening up pathways to accurate ML-driven reactive MD simulations that are more efficient than their classical counterparts.

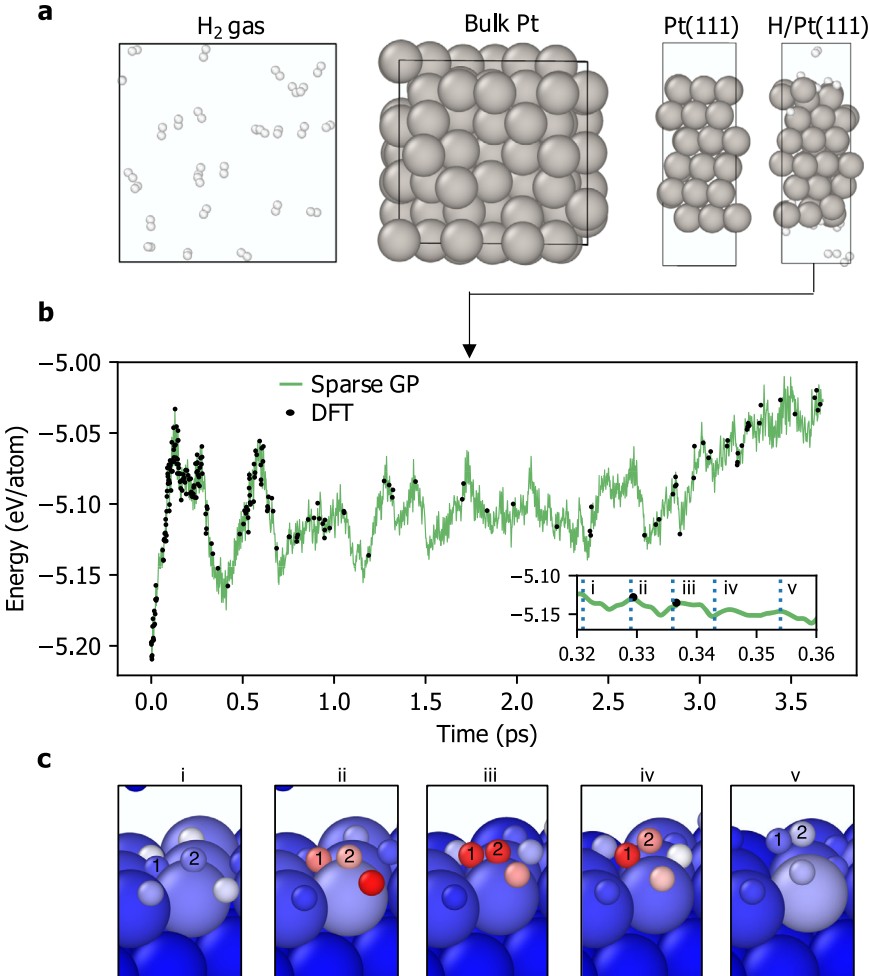

**Fig. 2 | On-the-fly training of the reactive SGP force field for H/Pt. a** Snapshots of four independent training simulations (left to right): gas-phase $H_2$; bulk Pt; (111) facet of the Pt surface; and $H_2$ interaction with the Pt(111) slab. **b** Energy versus time during the H/Pt(111) training simulation, with the SGP predictions in green and DFT evaluations shown as black dots. The inset is a zoom-in of the model predictions during the first recombination event observed at $t \approx 0.3$ ps. **c** Snapshots of the first recombination event. Atoms are colored by the uncertainty of the local energy prediction, with red corresponding to the prediction threshold $\Delta_{DFT}$.

## On-the-fly training of a reactive H/Pt force field

Figure 2 presents our procedure for training a reactive H/Pt force field on the fly. We performed four independent training simulations targeting gas-phase $H_2$, bulk and surface Pt, and $H_2$ interaction with the Pt(111) slab (Fig. 2a). Each training was performed from scratch, without any data initially stored in the training set of the SGP. As reported in Table 1, the single-element training simulations were each completed in less than two hours of wall time on 32 CPUs, with 24 DFT calls

**Table 1 | Summary of on-the-fly training for the H/Pt system: the simulation temperature $T$ (K); the total simulation time $\tau_{sim}$ (ps); the total wall time of the simulation $\tau_{wall}$ (h); the number of atoms in the simulation $N_{atoms}$; the total number of structures added to the training set $N_{struc}$; the total number of local environments in all training structures $N_{envs}$; the total number of local environments added to the sparse set $N_{sparse}$; and the total number of training labels $N_{labels}$**

| System | $T$ | $\tau_{sim}$ | $\tau_{wall}$ | $N_{atoms}$ | $N_{struc}$ | $N_{envs}$ | $N_{sparse}$ | $N_{labels}$ |
|---|---|---|---|---|---|---|---|---|
| $H_2$ | 1500 | 5.0 | 1.2 | 54 | 24 | 1296 | 124 | 4056 |
| Pt(111) | 300 | 4.0 | 1.2 | 54 | 4 | 216 | 87 | 676 |
| Pt | 1500 | 10.0 | 1.7 | 108 | 6 | 648 | 179 | 1986 |
| H/Pt | 1500 | 3.7 | 61.4 | 73 | 216 | 15 768 | 2034 | 48 816 |
| Total | - | - | 65.5 | - | 250 | 17 928 | 2424 | 55 534 |

made during the gas-phase $H_2$ simulation and only 4 and 6 calls made during the surface and bulk Pt simulations, respectively. The majority of DFT calls were made in the reactive H/Pt(111) simulation, with 216 calls made in total during the 3.7 ps simulation, resulting in nearly 50,000 training labels in total (consisting of the potential energy, atomic forces, and six independent stress tensor components from each DFT calculation). The order-of-magnitude increase in the number of accumulated sparse environments during the H/Pt training simulation reflects the greater diversity of chemical environments present in this two-phase simulation, encompassing gaseous $H_2$, H surface and sub-surface diffusion, and Pt surface and bulk dynamics. Potential energy predictions made during the run are plotted in Fig. 2b, showing excellent agreement between the SGP and DFT to within 1 meV/atom (see Supplementary Fig. 2 for the corresponding plots of the single-element training simulations).

The H/Pt(111) training was initialized with five randomly oriented $H_2$ molecules in the gas phase and with one side of the slab at full H coverage. The temperature was set at 1500 K to facilitate sampling of rare recombinative desorption events on the surface. The first recombination event occurred at $t \approx 0.3$ ps, shown as a sequence of MD snapshots in Fig. 2c. Here, each atom is colored by the uncertainty of its local energy, ranging from blue for negligible uncertainty to red corresponding to uncertainty near the prediction threshold $\Delta_{DFT}$. The formation of the H–H bond triggers two DFT calls (frames ii and iii), demonstrating the model's ability to automatically detect and

incorporate novel atomic environments into the training set without any prior information about the nature of the reaction.

### Force field model validation and comparisons with ReaxFF

We pool together all the structures and sparse environments collected in the four independent training simulations to construct the final SGP model, which we validate extensively on a range of properties against DFT. Our objective is to obtain a model that achieves accurate prediction of not only energy, forces, and stresses during MD simulations −quantities that the model was directly trained on−but also fundamental properties of Pt, $H_2$, and H/Pt that were not explicitly included in the training set. For bulk Pt, we predict the lattice constant to within 0.1% of the DFT value, as well as the bulk modulus and elastic constants to within 6% (see Table 2). The latter considerably improves on the recent ReaxFF force field for H/Pt[45], which overestimates the $C_{44}$ elastic constant by nearly 200%.

We also consider a more stringent test of model performance by forcing the model to extrapolate on structures that are significantly different from those encountered during training. In Fig. 3, we plot model predictions of bulk Pt energies as a function of volume, gas-phase $H_2$ dissociation and dimer interaction profiles, surface energies of several Pt facets, and H adsorption energies at different binding sites. In each case, we present the 99% confidence region associated with each prediction, which we compute under the Deterministic Training Conditional approximation of the GP predictive variance (see Methods).

In general, we observe low uncertainties and excellent agreement with DFT for configurations that are well-represented in the training set. For bulk Pt, the uncertainties nearly vanish close to the equilibrium volume (Fig. 3a), which was extensively sampled in the 0 GPa training simulation of bulk Pt. The model also gives confident and accurate predictions for $H_2$ bond lengths between ∼ 0.5 and 1.2 Å (Fig. 3b) and for dimer separations above 1.8 Å (Fig. 3c). The confidence region expectedly grows for extreme bond lengths and dimer separations that were not encountered during training (see Supplementary Fig. 3 for the radial distribution function of the $H_2$ training simulation averaged over all frames). For surface energies and H adsorption energies (Fig. 3d, e), the model agrees well with DFT for Pt(111). Surprisingly, the model is able to generalize and provide robust predictions for other surface facets that have not been included in training. The largest uncertainties are observed for H adsorption energies at (110) and (100) hollow sites, most likely due to geometric differences from the (111) binding site configurations.

Next, we validate model predictions of potential energies, atomic forces, and virial stress tensor components against DFT and compare with the ReaxFF model[45] (see Supplementary Figs. 7–9 for the parity plots). The models are tested on 50 regularly spaced frames from two 500-ps trajectories that were generated independently at 1200 K using the SGP and ReaxFF models. The structure was initialized with one side of the slab at full H coverage. We first highlight that qualitatively different behaviors are generated by the two force field models. While our

**Table 2 | Properties of bulk Pt, compared with experimental measurements and calculations using DFT, the SGP model, and the ReaxFF model[45]: the lattice constant $a$ (Å); the bulk modulus $B$ (GPa); and the elastic constants $C_{11}$, $C_{12}$, and $C_{44}$ (GPa)**

| Property | Experiment | DFT | SGP | ReaxFF |
|---|---|---|---|---|
| $a$ | 3.923 | 3.968 | 3.970 (0.1) | 3.947 (0.6) |
| $B$ | 280.1 | 248.0 | 258.2 (4.1) | 240.4 (−3.1) |
| $C_{11}$ | 348.7 | 314.5 | 323.2 (2.8) | 332.0 (5.6) |
| $C_{12}$ | 245.8 | 214.8 | 225.7 (5.1) | 194.6 (−9.4) |
| $C_{44}$ | 73.4 | 64.9 | 68.9 (6.2) | 194.6 (199.8) |

Percent errors relative to DFT are reported in parentheses for the SGP and ReaxFF models.

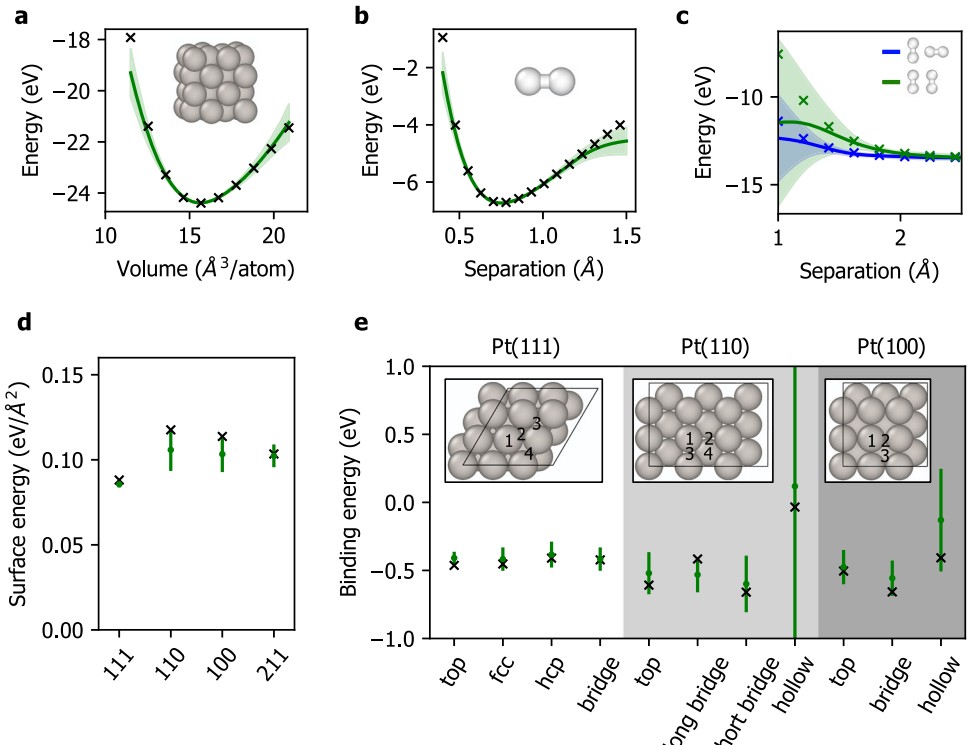

**Fig. 3 | Validation of the H/Pt SGP force field.** In each plot, shaded regions indicate 99% confidence regions of the SGP model. **a** Energy versus volume of bulk Pt. **b** Energy versus H–H bond distance of a single $H_2$ molecule. **c** Energy versus intermolecular distance of two $H_2$ molecules oriented perpendicular (blue) and parallel (green) to each other. **d** Surface energies of Pt for (111), (110), (100), and (211) facets. **e** H adsorption energies at different binding sites and Pt facets.

**Table 3 | H adsorption energies ($E_{ads}$) and diffusion barriers ($E_{dif}$) on Pt(111), compared with DFT and the SGP model**

| Quantity | Sites | DFT | | SGP | |
|---|---|---|---|---|---|
| $E_{ads}$ (eV) | FCC hollow | −0.52 | | −0.48 | |
| | HCP hollow | −0.47 | | −0.43 | |
| | Top | −0.49 | | −0.44 | |
| $E_{dif}$ (eV) | FCC ↔ HCP | 0.07 | 0.02 | 0.05 | 0.01 |
| | FCC ↔ Top | 0.15 | 0.12 | 0.15 | 0.10 |
| | HCP ↔ Top | 0.09 | 0.11 | 0.12 | 0.11 |

Forward and reverse barriers are listed for each diffusion pathway. See Supplementary Figs. 10 and 11 for the corresponding transition state pathways.

SGP model displays only adsorption and desorption events, the ReaxFF model gives rise to non-physical surface sublimation of isolated Pt hydride species, even at temperatures as low as 600 K (see Supplementary Fig. 14). To assess the fidelity of both models to DFT, we examine the accuracy of SGP and ReaxFF predictions on structures drawn from both MD trajectories.

Supplementary Table 1 summarizes the mean absolute errors (MAE) with respect to DFT for the three classes of properties. Overall, the SGP predictions demonstrate significantly higher fidelity to DFT than the ReaxFF predictions for both trajectories. In particular, even though the surface-evaporated structures encountered in the ReaxFF trajectory were not included in our training set, the SGP model is capable of extrapolating on these structures closer to the DFT values than the ReaxFF model. This improvement is most apparent in the potential energy predictions on the ReaxFF trajectory. ReaxFF predicts the high-energy evaporated structures to be low-energy (MAE = 93 meV/atom); the SGP model does so as well, but to a much lesser extent (MAE = 26 meV/atom), as evidenced by the cluster of data points being closer to the parity line in Supplementary Fig. 8.

We now examine full transition state pathways for $H_2$ dissociative adsorption and atomic H diffusion on Pt(111) at the low-coverage limit (Supplementary Fig. 11). Both DFT and our SGP model predict the three binding sites (FCC hollow, HCP hollow, top) to be nearly isoenergetic (within ~0.05 eV of each other), with the FCC hollow site being the most stable (Table 3). In contrast, ReaxFF predicts the top site to be the most stable (by 0.16 eV compared to the HCP hollow site) and as such it was precluded from the analysis here. According to the minimum energy pathway obtained from DFT (Supplementary Fig. 10), gas-phase $H_2$ molecule first approaches the top site, followed by dissociation into the nearest FCC hollow sites. Molecular adsorption is not favored, and as such the structure spontaneously relaxes to the fully dissociated state. The chemisorption process is barrierless and overall exothermic by 1.02 eV. Subsequent diffusion from FCC to HCP hollow site has the smallest energy barrier (0.07 eV), followed by HCP hollow to top (0.09 eV) and FCC hollow to top (0.15 eV). As summarized in Table 3, the SGP predictions of these barriers (and the associated pathways; see Supplementary Fig. 10) are in excellent agreement with DFT.

Lastly, we examine atomic H diffusion using SGP to perform MD simulations in the temperature range of 300–900 K (for details see Supplementary Figs. 12–14 in SI). The Arrhenius analysis yields an apparent activation energy of 92 meV and a diffusion prefactor of $3.53 \times 10^{13}$ Å$^2$/s (Supplementary Fig. 13), in good agreement with the experimental values of 68 meV and $1 \times 10^{13}$ Å$^2$/s from helium atom scattering measurements[54]. This dynamical estimate of the diffusion barrier is consistent with the static values from the transition state pathways (Table 3), in the range of 50–150 meV (SGP) and 65-149 meV (DFT).

#### Large-scale reactive MD simulations

The trained SGP model was mapped onto a fast polynomial model, as described in Methods, and used to perform explicit two-phase reactive

MD simulations of $H_2$ turnover on Pt(111). These large-scale simulations allow for a direct estimation of the reaction rates of $H_2$ dissociative adsorption and recombinative desorption, resembling a realistic experimental measurement. The system was initialized with both sides of the slab at full H coverage and 80 randomly oriented $H_2$ molecules in the gas phase, giving 864 Pt atoms and 448 H atoms in total (see Fig. 4a for an example MD snapshot). The simulations were performed at four temperatures—450, 600, 750, and 900 K—with the vertical dimension of the box frozen at 6.8 nm and with the pressure along the transverse axes set to 0 GPa. 500 ps of dynamics were simulated in total with a time-step of 0.1 fs. Reactive events were counted by monitoring the coordination number of each H atom within a 1 Å shell, which is zero for atomic H adsorbed on the surface, and one for molecular $H_2$ in the gas phase.

During the first ~100–200 ps, the recombination rate exceeds the dissociation rate as the system approaches an equilibrium surface H coverage. Higher temperatures are seen to be associated with lower equilibrium coverage values (see Supplementary Fig. 5 for a plot of the surface coverage as a function of time). Once the slab reaches an equilibrium coverage, the rates of dissociation and recombination become roughly equal (Fig. 4b). The reaction rates are estimated by performing a linear fit of the cumulative number of reactive events for the final 300 ps of each simulation. The Arrhenius plot of these rates against inverse temperature provides an apparent activation energy of 0.25(2) eV (Fig. 4c). To check the effect of pressure, the simulations were repeated with double the size of the vacuum, giving a consistent estimate of 0.20(3)eV for the apparent activation energy (Supplementary Fig. 6). Both estimates are in good agreement with the experimental values reported in the range of 0.21–0.24 eV from surface science experiments conducted at ultrahigh vacuum[55], as well as ambient pressures[56].

## Discussion

We have developed an active learning method for autonomous, on-the-fly training of reactive force fields, achieving excellent accuracy relative to DFT and computational efficiency surpassing that of the primary traditional force field used for reactive simulations, ReaxFF. The method presented here is intended to reduce the time, effort, and expertise required to train accurate reactive force fields, making it possible to obtain a high-quality model in a matter of days with minimal human supervision. In particular, our method has enabled direct and unbiased MD simulations of $H_2$ turnover on Pt(111) including all degrees of freedom with explicit molecular gas phase and chemisorbed H atoms, thereby overcoming the limitations of previous studies constrained to either an implicit gas phase or a single $H_2$ molecule.

Our method bridges two of the leading approaches to ML force fields, combining the principled Bayesian uncertainties available in kernel-based approaches like GAP with the computational efficiency of parametric force fields such as SNAP[11], qSNAP[57], and MTP[12]. By simplifying the training procedure and reducing the computational cost of reactive force fields at the same time, this unified approach will help extend the range of applicability and predictive power of reactive MD as a modeling tool, providing new insights into complex systems that have so far remained out of reach. Identifying more compact descriptors for many-element systems and distributing automated training protocols across many processors are promising directions that would help extend the method presented here to systems of even greater complexity. Of particular interest are applications to biochemical reactions and more complex heterogeneous catalysis.

## Methods

Here, we present our implementation of SGP force fields, our procedure for mapping them onto accelerated polynomial models, and the computational details of our training workflow, DFT calculations, MD simulations, and transition state modeling.

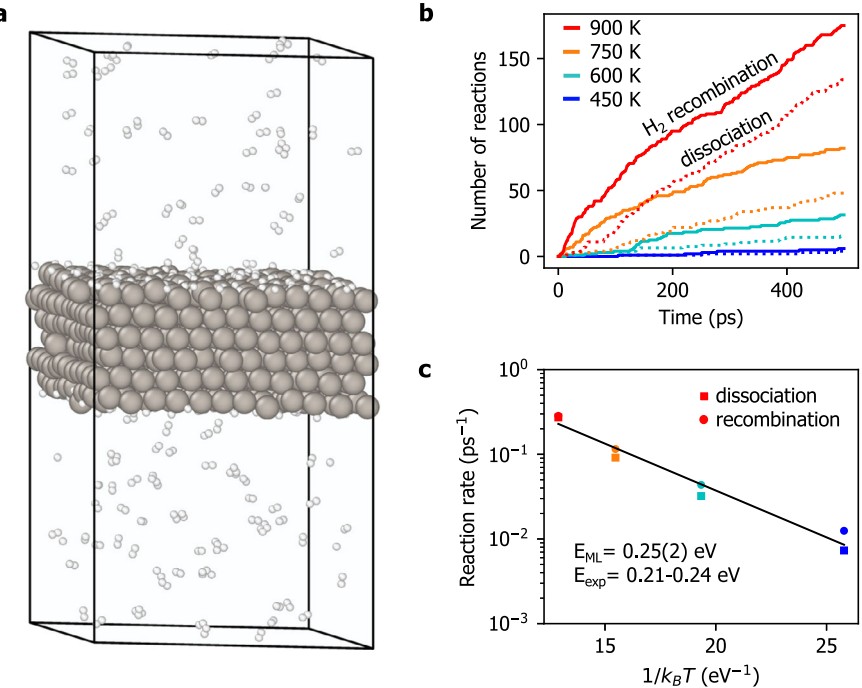

**Fig. 4 | Reactive MD simulation of H/Pt(111) with the mapped SGP force field.** **a** Snapshot from a 500 ps simulation at 900 K. The system is a six-layer slab model of a 12 × 12 unit cell of Pt(111) with $H_2$ reactive events occurring at both sides of the slab. **b** Cumulative number of dissociation and recombination events observed at 450 K (dark blue), 600 K (light blue), 750 K (orange), and 900 K (red). The reaction rate is estimated from the slope of each curve after an initial equilibration period of 200ps. **c** Arrhenius plot of the reaction rate versus inverse temperature. An apparent activation energy of 0.25(2) eV is obtained, in good agreement with the experimental values of 0.21–0.24 eV[55,56].

## Sparse Gaussian process (SGP) force fields

Our SGP force fields are defined in three key steps: (i) Fixing the energy model, which involves expressing the total potential energy as a sum over local energies assigned to local atomic environments $\rho_i$, (ii) mapping these local environments onto descriptor vectors $\mathbf{d}_i$ that serve as input to the model, and (iii) choosing a kernel function $k(\mathbf{d}_i, \mathbf{d}_j)$ that quantifies the similarity between two local environments. In this section, we summarize these steps and then present our approach to making predictions, evaluating Bayesian uncertainties, and optimizing hyperparameters with the SGP.

**Defining the energy model.** As in the Gaussian Approximation Potential formalism[10], we express the total potential energy $E(\mathbf{r}_1, ..., \mathbf{r}_N; s_1, ... s_N)$ of a structure of $N$ atoms as a sum over local energies $\varepsilon$ assigned to atom-centered local environments,

$$E(\mathbf{r}_1, ..., \mathbf{r}_N; s_1, ... s_N) = \sum_i^N \varepsilon(s_i, \rho_i). \quad (1)$$

The decomposition of the total energy into a sum over local, atom-centered contributions gives the model the key property of scaling linearly with the number of atoms in the system. Here $\mathbf{r}_i$ is the position of atom $i$, $s_i$ is its chemical species, and we define the local environment $\rho_i$ of atom $i$ to be the set of chemical species and interatomic distance vectors connecting atom $i$ to neighboring atoms $j \neq i$ within a cutoff sphere of radius $r_{\text{cut}}^{(s_i, s_j)}$,

$$\rho_i = \left\{ (s_j, \mathbf{r}_{ij}) | r_{ij} < r_{\text{cut}}^{(s_i, s_j)} \right\}. \quad (2)$$

Note that we allow the cutoff radius $r_{\text{cut}}^{(s_i, s_j)}$ to depend on the central and environment species $s_i$ and $s_j$. This additional flexibility was found to be important in the H/Pt system, with H–H and H–Pt interactions requiring shorter cutoffs than Pt-Pt.

**Describing local environments.** To train an SGP model, local environments $\rho_i$ must be mapped onto fixed-dimension descriptor vectors $\mathbf{d}_i$ that respect the physical symmetries of the potential energy surface. Different from SOAP descriptors used in GAP, we use the multielement atomic cluster expansion of Drautz[50,58] to efficiently compute many-body descriptors that satisfy rotational, permutational, translational, and mirror symmetry. To satisfy rotational invariance, the descriptors are computed in a two-step process: the first step is to compute a rotationally equivariant descriptor $\mathbf{c}_i$ of atom $i$ by looping over the neighbors of $i$ and passing each interatomic distance vector $\mathbf{r}_{ij}$ through a basis set of radial basis functions multiplied by spherical harmonics, and the second step is to transform $\mathbf{c}_i$ into a rotationally invariant descriptor $\mathbf{d}_i$ by forming a tensor product of $\mathbf{c}_i$ with itself and keeping only the rotationally invariant elements of the resulting tensor. In the first step, the interatomic distance vectors $\mathbf{r}_{ij}$ are passed through basis functions $\phi_{n\ell m}$ of the form

$$\phi_{n\ell m}^{(s_i, s_j)}(\mathbf{r}_{ij}) = R_n(\tilde{r}_{ij}) Y_{\ell m}(\hat{r}_{ij}) c\left( r_{ij}, r_{\text{cut}}^{(s_i, s_j)} \right), \quad (3)$$

where $\tilde{r}_{ij} = \frac{r_{ij}}{r_{\text{cut}}^{(s_i, s_j)}}$ is a scaled interatomic distance, $R_n$ are radial basis functions defined on the interval $[0, 1]$, $Y_{\ell m}$ are the real spherical harmonics, and $c$ is a cutoff function that smoothly goes to zero as the interatomic distance $r_{ij}$ approaches the cutoff radius $r_{\text{cut}}^{(s_i, s_j)}$. The equivariant descriptor $\mathbf{c}_i$ (called the "atomic base" in ref. 50) is a tensor indexed by species $s$, radial number $n$ and angular numbers $\ell$ and $m$, and is computed by summing the basis functions over all neighboring atoms of a particular species $s$,

$$c_{isnlm} = \sum_{j \in \rho_i} \delta_{s, s_j} \phi_{n\ell m}^{(s_i, s_j)}(\mathbf{r}_{ij}), \quad (4)$$

where $\delta_{s, s_j} = 1$ if $s = s_j$ and 0 otherwise. Finally, in the second step, the rotationally invariant vector $\mathbf{d}_i$ is computed by invoking the sum rule of

spherical harmonics,

$$d_{i s_1 s_2 n_1 n_2 \ell} = \sum_{m=-\ell}^{\ell} c_{i s_1 n_1 \ell m} c_{i s_2 n_2 \ell m}. \qquad (5)$$

To eliminate redundancy in the invariant descriptor, we notice that interchanging the $s$ and $n$ indices leaves the descriptor unchanged,

$$d_{i s_1 s_2 n_1 n_2 \ell} = d_{i s_2 s_1 n_2 n_1 \ell}. \qquad (6)$$

In practice we keep only the unique values, which can be visualized as the upper- or lower-triangular portion of the matrix formed from invariant contractions of the tensor product $\mathbf{c}_i \otimes \mathbf{c}_i$, as shown schematically in Fig. 1(b). This gives a descriptor vector of dimension

$$n_{\text{desc}} = \frac{N_{\text{species}} N_{\text{rad}} (N_{\text{species}} N_{\text{rad}} + 1)(\ell_{\max} + 1)}{2}, \qquad (7)$$

where $N_{\text{species}}$ is the number of species, $N_{\text{rad}}$ is the number of radial functions, and $\ell_{\max}$ is the maximum value of $\ell$ in the spherical harmonics expansion. When computing descriptor values, we also compute gradients with respect to the Cartesian coordinates of each neighbor in the local environment, which are needed to evaluate forces and stresses.

The cutoff radii $r_{\text{cut}}^{(s_i, s_j)}$ and radial and angular basis set sizes $N_{\text{rad}}$ and $\ell_{\max}$ are hyperparameters of the model that can be tuned to improve accuracy. We chose the Chebyshev polynomials of the first kind for the radial basis set and a simple quadratic for the cutoff,

$$c\left(r_{ij}, r_{\text{cut}}^{(s_i, s_j)}\right) = \left(r_{\text{cut}}^{(s_i, s_j)} - r_{ij}\right)^2. \qquad (8)$$

We set the Pt-Pt cutoff to 4.25 Å and the H−H and H−Pt cutoffs to 3.0 Å, and truncated the basis expansion at $N_{\text{rad}} = 8$, $\ell_{\max} = 3$. These hyperparameters performed favorably on a restricted training set of H/Pt structures, as determined by the log marginal likelihood of the SGP (see Supplementary Fig. 1).

**Making model predictions.** The SGP prediction of the local energy $\varepsilon$ assigned to environment $\rho_i$ is evaluated by performing a weighted sum of kernels between $\rho_i$ and a set of representative sparse environments $S$,

$$\varepsilon(\rho_i) = \sum_{t \in S}^{N_S} k(\mathbf{d}_i, \mathbf{d}_t) \alpha_t, \qquad (9)$$

where $N_S$ is the number of sparse environments, $k$ is a kernel function quantifying the similarity of two local environments, and $\boldsymbol{\alpha}$ is a vector of training coefficients. For the kernel function, we use a normalized dot product kernel raised to an integer power $\xi$, similar to the SOAP kernel[51]:

$$k(\mathbf{d}_1, \mathbf{d}_2) = \sigma^2 \left(\frac{\mathbf{d}_1 \cdot \mathbf{d}_2}{d_1 d_2}\right)^{\xi}. \qquad (10)$$

Here, $d_1 = \|\mathbf{d}_1\|$ and $d_2 = \|\mathbf{d}_2\|$. The hyperparameter $\sigma$ quantifies variation in the learned local energy, and in our final trained model is set to 3.84 eV.

The training coefficients $\boldsymbol{\alpha}$ are given by

$$\boldsymbol{\alpha} = \boldsymbol{\Sigma} \mathbf{K}_{SF} \mathbf{y}, \qquad (11)$$

where $\boldsymbol{\Sigma} = (\mathbf{K}_{SF} \boldsymbol{\Lambda}^{-1} \mathbf{K}_{FS} + \mathbf{K}_{SS})^{-1}$, $\mathbf{K}_{SF}$ is the matrix of kernel values between the sparse set $S$ and the training set $F$ (with $\mathbf{K}_{FS} = \mathbf{K}_{SF}^{\top}$), $\mathbf{K}_{SS}$ is

the matrix of kernel values between the sparse set $S$ and itself, $\boldsymbol{\Lambda}$ is a diagonal matrix of noise values quantifying the expected error associated with each training label, and $\mathbf{y}$ is the vector of training labels consisting of potential energies, forces, and virial stresses. For the noise values in $\boldsymbol{\Lambda}$, we chose for the final trained model a force noise of $\sigma_F = 0.1$ eV/Å, an energy noise of $\sigma_E = 50$ meV, and a stress noise of $\sigma_S = 0.1$ GPa. In practice, we found that performing direct matrix inversion to compute $\boldsymbol{\Sigma}$ in Eq. (11) was numerically unstable, so we instead compute $\boldsymbol{\alpha}$ with QR decomposition, as proposed in ref. 59.

**Evaluating uncertainties.** To evaluate uncertainties on total potential energies $E$, we compute the GP predictive variance $V_E$ under the Deterministic Training Conditional (DTC) approximation[60],

$$V_E = k_{EE} - \mathbf{k}_{ES} \mathbf{K}_{SS}^{-1} \mathbf{k}_{ES} + \mathbf{k}_{ES} \boldsymbol{\Sigma} \mathbf{k}_{SE}. \qquad (12)$$

Here $k_{EE}$ is the GP covariance between $E$ and itself, which is computed as a sum of local energy kernels

$$k_{EE} = \langle E, E \rangle = \sum_{i,j=1}^{N} \langle \varepsilon_i, \varepsilon_j \rangle = \sum_{i,j=1}^{N} k(\mathbf{d}_i, \mathbf{d}_j) \qquad (13)$$

with $i$ and $j$ ranging over all atoms in the structure. The row vector $\mathbf{k}_{ES}$ stores the GP covariances between the potential energy $E$ and the local energies of the sparse environments, with $\mathbf{k}_{SE} = \mathbf{k}_{ES}^{\top}$.

Surface energies and binding energies are linear combinations of potential energies, and their uncertainties can be obtained from a straightforward generalization of Eq. (12). Consider a quantity $Q$ of the form $Q = aE_1 + bE_2$, where $a$ and $b$ are scalars and $E_1$ and $E_2$ are potential energies. GP covariances are bilinear, so that for instance

$$\langle Q, E \rangle = a \langle E_1, E \rangle + b \langle E_2, E \rangle, \qquad (14)$$

and as a consequence the GP predictive variance assigned to $Q$ is obtained by replacing $k_{EE}$ and $\mathbf{k}_{ES}$ in Eq. (12) with $k_{QQ}$ and $\mathbf{k}_{QS}$, respectively, where

$$k_{QQ} = \langle Q, Q \rangle = a^2 k_{E_1 E_1} + b^2 k_{E_2 E_2} + 2ab k_{E_1 E_2} \qquad (15)$$

and

$$\mathbf{k}_{QS} = a \mathbf{k}_{E_1 S} + b \mathbf{k}_{E_2 S}. \qquad (16)$$

We use these expressions to assign confidence regions to the surface and binding energies reported in Fig. 3.

To evaluate uncertainties on local energies $\varepsilon$, we first compute a simplified predictive variance

$$V_\varepsilon = k_{\varepsilon\varepsilon} - \mathbf{k}_{\varepsilon S} \mathbf{K}_{SS}^{-1} \mathbf{k}_{S\varepsilon}. \qquad (17)$$

Formally, $V_\varepsilon$ is the predictive variance of an exact GP trained on the local energies of the sparse environments, and it has two convenient properties: (i) it is independent of the noise hyperparameters $\boldsymbol{\Lambda}$, and (ii) it is proportional to but otherwise independent of the signal variance $\sigma^2$. This allows us to rescale the variance to obtain a unitless measure of uncertainty $\widetilde{V}_\varepsilon$,

$$\widetilde{V}_\varepsilon = \frac{1}{\sigma^2} V_\varepsilon. \qquad (18)$$

Notice that $\widetilde{V}_\varepsilon$ lies between 0 and 1 and is independent of the kernel hyperparameters, providing a robust uncertainty measure on local environments that we use to guide our active learning protocol.

**Optimizing hyperparameters.** To optimize the hyperparameters of the SGP, we evaluate the DTC log marginal likelihood

$$\mathcal{L} = -\frac{1}{2}\log|\mathbf{K}_{SF}\mathbf{K}_{SS}^{-1}\mathbf{K}_{FS} + \mathbf{\Lambda}| - \frac{1}{2}\mathbf{y}^{\top}(\mathbf{K}_{SF}\mathbf{K}_{SS}^{-1}\mathbf{K}_{FS} + \mathbf{\Lambda})^{-1}\mathbf{y} - \frac{N_{\text{labels}}}{2}\log(2\pi),$$
(19)

where $N_{\text{labels}}$ is the total number of lables in the training set. Eq. (19) quantifies the likelihood of the training labels $\mathbf{y}$ given a particular choice of hyperparameters. The first term penalizes model complexity while the second measures the quality of the fit, and hence hyperparameters that maximize $\mathcal{L}$ tend to achieve a favorable balance of complexity and accuracy[61]. During on-the-fly runs, after each of the first ten updates to the SGP, the kernel hyperparameters $\sigma$, $\sigma_E$, $\sigma_F$, and $\sigma_S$ are optimized with the L-BFGS algorithm by evaluating the gradient of $\mathcal{L}$. We also use the log marginal likelihood to evaluate different descriptor hyperparameters $N_{\text{rad}}$, $\ell_{\max}$, and $r_{\text{cut}}^{(s_i, s_j)}$ and the discrete kernel hyperparameter $\xi$ (see Supplementary Fig. 1 for a comparison).

## Mapping to an equivalent polynomial model

Letting $\tilde{\boldsymbol{d}}_i = \frac{\mathbf{d}_i}{d_i}$ denote the normalized descriptor of local environment $\rho_i$, we observe that with the dot product kernel defined in Eq. (10), local energy prediction can be rewritten as

$$
\begin{aligned}
\varepsilon(\rho_i) &= \sigma^2 \sum_t (\tilde{\boldsymbol{d}}_i \cdot \tilde{\boldsymbol{d}}_t)^{\xi} \alpha_t \\
&= \sigma^2 \sum_{t, m_1, \ldots, m_\xi} \tilde{d}_{im_1}\tilde{d}_{tm_1} \cdots \tilde{d}_{im_\xi}\tilde{d}_{tm_\xi} \alpha_t \\
&= \sigma^2 \sum_{m_1, \ldots, m_\xi} \tilde{d}_{im_1} \cdots \tilde{d}_{im_\xi} \left( \sum_t \tilde{d}_{tm_1} \cdots \tilde{d}_{tm_\xi} \alpha_t \right) \\
&= \sum_{m_1, \ldots, m_\xi} \tilde{d}_{im_1} \cdots \tilde{d}_{im_\xi} \beta_{m_1, \ldots, m_\xi},
\end{aligned}
$$
(20)

where in the final two lines we have gathered all terms involving the sparse set into a symmetric tensor $\boldsymbol{\beta}$ of rank $\xi$. Once $\boldsymbol{\beta}$ is computed, mean predictions of the SGP can be evaluated without performing a loop over sparse points, which can considerably accelerate model predictions for small $\xi$. For $\xi = 1$, corresponding to a simple dot product kernel, mean predictions become linear in the descriptor,

$$\varepsilon_{\xi=1}(\rho_i) = \tilde{\boldsymbol{d}} \cdot \boldsymbol{\beta}.$$
(21)

Evaluating the local energy with Eq. (21) requires a single dot product rather than a dot product for each sparse environment, accelerating local energy prediction with the SGP by a factor of $N_S$. For $\xi = 2$, mean predictions are quadratic in the descriptor and can be evaluated with a vector-matrix-vector product,

$$\varepsilon_{\xi=2}(\rho_i) = \tilde{\boldsymbol{d}}^{\top} \boldsymbol{\beta} \tilde{\boldsymbol{d}}.$$
(22)

The cost of the matrix-vector product $\boldsymbol{\beta}\tilde{\boldsymbol{d}}$ scales quadratically with the descriptor dimension $n_{\text{desc}}$ and becomes the principal bottleneck when the descriptor dimension is large, as shown in Supplementary Figs. 15 and 16.

The mapping in Eq. (22) is exact and provides an acceleration of SGP local energy prediction if the number of sparse environments exceeds the descriptor dimension, $N_S > n_{\text{desc}}$, with quadratic prediction expected to be faster by a factor of $\frac{N_S}{n_{\text{desc}}}$. For general $\xi$, this ratio of efficiencies is equal to $\frac{N_S}{n_{\text{desc}}^{\xi}}$ and diminishes rapidly with $\xi$. In our H/Pt models, for which $N_S = 2424$ and $n_{\text{desc}} = 544$, we found $\xi = 2$ models to give considerable improvement over $\xi = 1$, but found no benefit for $\xi \geq 3$ (see Supplementary Table I). We therefore selected $\xi = 2$ for our final trained model and used quadratic prediction when performing

production MD simulations, giving a theoretical speed up over SGP mean prediction of $\frac{N_S}{n_{\text{desc}}} \approx 4.5$. In practice we observe a speed up of greater than 10x due to better optimization of the quadratic LAMMPS model (see Supplementary Fig. 4).

## Computational details

**Training workflow.** To calculate the ACE descriptors, train SGP models, and map SGPs onto accelerated quadratic models, we have developed the FLARE++ code, available at https://github.com/mir-group/flare_pp. On-the-fly training simulations were performed with the FLARE code[32], available at https://github.com/mir-group/flare, which is coupled to the MD engine as implemented in the Atomic Simulation Environment (ASE)[62] and the DFT engine as implemented in the Vienna Ab Initio Simulation Package (VASP)[63].

**Density functional theory (DFT).** We perform DFT calculations using plane-wave basis sets and the projector augmented-wave (PAW) method as implemented in the Vienna Ab Initio Simulation Package (VASP)[63]. The plane-wave kinetic energy cutoff is set at 450 eV. The Methfessel-Paxton smearing scheme is employed with a broadening value of 0.2 eV. The total energy is converged to $10^{-5}$ eV. Gas-phase $H_2$ is optimized in a $14 \times 15 \times 16$ Å$^3$ cell at the $\Gamma$-point. The lattice constant of bulk face-centered cubic Pt is optimized according to the third-order Birch-Murnaghan equation of state, using a $19 \times 19 \times 19$ $k$-point grid. For training and benchmark, we use a six-layer slab model of a $3 \times 3$ unit cell of Pt(111). The slab is spaced with 16 Å of vacuum along the direction normal to the surface in order to avoid spurious interactions between adjacent unit cells. The Brillouin zone is sampled using a $\Gamma$-centered $7 \times 7 \times 1$ $k$-point grid. For force field validation (Supplementary Figs. 7–9), we use a slightly larger $5 \times 5$ unit cell with a $5 \times 5 \times 1$ $k$-point grid.

We employ the Perdew-Burke-Ernzerhof (PBE) parametrization[64] of the generalized gradient approximation (GGA) of the exchange-correlation functional. PBE provides Pt lattice constant of 3.97 Å, in good agreement with the experimental benchmark of 3.92 Å[65]. We also examine the dissociative adsorption energy of $H_2$, defined as the change in energy from an isolated slab and a gas phase $H_2$ to a combined system of atomic H adsorbed on the slab:

$$E_{\text{ads}} = E_{H_{(\text{ads})}/Pt(111)} - E_{Pt(111)} - \frac{1}{2}E_{H_{2_{(g)}}}.$$
(23)

PBE provides H adsorption energy of $-0.52$ eV at the dilute limit, within 0.16 eV of the experimental benchmark of $-0.36$ eV[66]. Based on these observations, we conclude that PBE is an appropriate reference functional that can provide reasonable comparison with experiments for $H_2$ chemisorption on Pt(111)[67].

**Molecular dynamics (MD).** We perform MD simulations using the Large-scale Atomic/Molecular Massively Parallel Simulator (LAMMPS)[68] with our custom pair-style for mapped SGP force fields available in the FLARE++ repository. We also make various comparisons with an available H/Pt ReaxFF model[45,69] for force field validation (Supplementary Figs. 7–9 and 14). For simulations of $H_2$ reactivity, we use a six-layer slab model of a $12 \times 12$ unit cell of Pt(111). Periodic boundary condition is enforced in all Cartesian directions. The box dimension normal to the surface is fixed to retain the vacuum. A velocity-Verlet integrator is used with a time-step of $\delta t = 0.1$ fs to evolve the equations of motion. Production simulations are run for 500 ps within the isothermal-isobaric (NPT) ensemble at a temperature range of 300–900 K. Pressure and temperature are enforced on the system using a Nosé-Hoover barostat (1000 $\delta t = 100$ fs coupling) and thermostat (100 $\delta t = 10$ fs coupling), respectively.

For Arrhenius analysis of atomic H diffusion (Supplementary Figs. 12 and 13), we use a $10 \times 10$ unit cell of Pt(111). The bottommost

layer is immobilized (velocities and forces set to zero) to prevent it from acting as a surface. All simulations consist of 10 ps equilibration within the isothermal-isobaric (NPT) ensemble, followed by 200 ps production within the canonical (NVT) ensemble at the same temperature range of 300–900 K. We take the second half of the production as a linear diffusive regime, where the mean-squared displacement (MSD) behaves linearly as a function of time. For each temperature, the MSD is averaged over 30 parallel replicas to ensure sufficient noise reduction.

**Transition state modeling.** We perform ab initio transition state modeling using the VASP Transition State Tools (VTST) (Supplementary Figs. 10 and 11). The bottom three layers are fixed at their bulk positions to mimic bulk properties. All initial and final states are optimized via ionic relaxation, with the total energy and forces converged to $10^{-5}$ eV and 0.02 eV/Å, respectively. Transition state pathways are first optimized via the climbing-image nudged elastic band (CI-NEB) method, using three intermediate images generated by linear interpolation with a spring constant of 5 eV/Å$^2$. The total forces, defined as the sum of the spring force along the chain and the true force orthogonal to the chain, are converged to 0.05 eV/Å. Then, the image with the highest energy is fully optimized to a first-order saddle point via the dimer method, this time converging the total energy and forces to $10^{-7}$ eV and 0.01 eV/Å, respectively. We confirm that the normal modes of all transition states contain only one imaginary frequency by calculating the Hessian matrix within the harmonic approximation, using central differences of 0.01 Å at the same level of accuracy as the dimer method.

For comparison against DFT, the transition state pathways are also optimized with our SGP model via the CI-NEB method as implemented in LAMMPS, this time using ten intermediate images generated by linear interpolation.

## Data availability
Input and output files of the molecular dynamics simulations described in this study are available at https://archive.materialscloud.org/record/2022.92.

## Code availability
An open-source implementation of the FLARE++ code is available at https://github.com/mir-group/flare_pp.

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

## Acknowledgements

We acknowledge enlightening discussions with Albert P. Bartók, Steven B. Torrisi, Lixin Sun, Matthew M. Montemore, and Philippe Sautet. We also thank David Clark, Anders Johansson, and Claudio Zeni for their contributions to the FLARE++ code. J.V. was supported by Bosch Research LLC and the National Science Foundation under Grant No. 1808162 and 2003725. Y.X. was supported by the US Department of Energy (DOE), Office of Science, Office of Basic Energy Sciences (BES) under Award No. DE-SC0020128 (Design & Assembly of Atomically Precise Quantum Materials & Devices). J.S.L. was supported by the Integrated Mesoscale Architectures for Sustainable Catalysis (IMASC), an Energy Frontier Research Center funded by the US Department of Energy (DOE), Office of Science, Office of Basic Energy Sciences (BES)

under Award No. DE-SC0012573. C.J.O. was supported by the National Science Foundation Graduate Research Fellowship Program under Grant No. DGE1745303. J.V., J.S.L., and C.J.O used the Cannon cluster, FAS Division of Science, Research Computing Group at Harvard University. J.S.L. and C.J.O. used the National Energy Research Scientific Computing Center (NERSC), a DOE Office of Science User Facility supported under Contract No. DE-AC02-05CH11231, through allocation m3275.

## Author contributions

J.V. designed the method, performed simulations, led development of the FLARE++ code, and implemented the mapping of the SGP model and the LAMMPS pair-style with contributions from Y.X. Y.X. implemented the on-the-fly training workflow coupled with ASE. J.S.L. performed force field validation, transition state modeling, and comparisons with ReaxFF. C.J.O. performed the Arrhenius analysis of atomic H diffusion. B.K. conceived the reactive application, contributed to algorithm development, and supervised the work. J.V. wrote the manuscript. All authors contributed to manuscript preparation.

## Competing interests

The authors declare no competing interests.
