## [Peer Review File · Nature Communications]

nature portfolio

Peer Review File

Draft OnlyREVIEWER COMMENTS

Reviewer #1 (Remarks to the Author):

In the manuscript "Active learning of reactive Bayesian force fields: Application to heterogeneous catalysis dynamics of H/Pt", the authors demonstrate a numerically efficient approach for approximating an ab initio potential energy surface with a machine learning model for molecular dynamics calculations. The key contribution is i) an end-to-end workflow requiring minimal human interaction and ii) a neat trick to replace a comparison of atomic environments within a polynomial kernel with a fixed tensor product. I suggest publication with minor aspects improved as outlined below.

With the promise of machine learning that more data yields more accuracy, it is refreshing to see work on rendering this process affordable and, hence, of practical use. While the key ingredients of the workflow are close to refs 29 and 32 (with some overlap in authors), the computational simplification is a noteworthy and useful result. Making the corresponding code available will certainly help the community to evaluate this approach. The point of the authors that their model is faster than ReaxFF is highly implementation dependent - can the authors comment on the formal scaling behaviour of their method with e.g. number of elements, increasing disorder, number of atoms? This would strengthen the argument and would guide the reader.

Considering Table 1, the interface itself seems to be dominating the training cost. How do the authors expect the required training data to behave in the "more complex heterogeneous catalysis" applications the discussion points out, e.g. with adding more surface features or surface size?

In Figs 3d and 3e the model seems to yield reasonable values for other facets, which is reassuring for somewhat similar local atomic environments, which would explain why the uncertainty for the hollow site for Pt(110) is so high. To me it therefore is counterintuitive that Fig 3b seems to show a substantial error and uncertainty for H2 distances $>1.3\text{\AA}$ that are likely to occur considering the cell in Fig 4a. Can the authors comment on why this did not trigger additional DFT calculations or why these distances are not in the training data?

Reviewer #2 (Remarks to the Author):

In this paper, Vandermause et al. describe the application of reactive Bayesian force-fields to H₂ dissociation and recombination on Pt(111). The paper is well written and quite interesting and thus certainly publishable in principle. I have some comments though:

1) I think that the methodological advances in this paper are not very significant. Unlike the previous FLARE model by the authors, a sparse GP model is used. But (as the authors note) this technology has been used in the GAP method for a long time. In fact, the current method is basically a reimplementaion of GAP with only rather small technical differences. Furthermore, the work of Kresse and co-workers (e.g. refs [33] and [34]) uses a very similar approach in the context of on-the-fly MD simulations. Although these papers are cited, I feel like these close relationships are not adequately acknowledged in the text.

2) The main novelty of the approach is the polynomial mapping, which allows accelerating the simulations once the training is completed. In this context, the authors write that "[e]valuating the local energy with Eq. (20) requires a single dot product rather than a dot product for each sparse environment, accelerating local energy prediction with the SGP by a factor of N_S . For $\xi = 2$, mean predictions are quadratic in the descriptor and can be evaluated with a vector-matrix-vector product". This sounds nice, but the acceleration by a factor N_S is only valid for the linear kernel, which the authors do not use and no statement about the acceleration for the quadratic kernel is made. More importantly, this comparison assumes that kernel evaluation rather than construction of the descriptor is the rate-limiting step for the MD simulation. In my experience, computing the descriptor and its derivatives takes a significant part of the computational effort though. The authors should therefore provide more robust data on how much the polynomial form actually accelerates the method.

3) As always with iterative training schemes, I wonder how the authors decided that the training was completed?

Active learning of reactive Bayesian force fields: Application to heterogeneous catalysis dynamics of H/Pt

Jonathan Vandermause^{1,2}, Yu Xie², Jin Soo Lim³, Cameron J. Owen³, and Boris Kozinsky^{2,4}

1 Department of Physics, Harvard University, Cambridge, MA 02138, USA

2 John A. Paulson School of Engineering and Applied Sciences, Harvard University, Cambridge, MA 02138, USA

3 Department of Chemistry and Chemical Biology, Harvard University, Cambridge, MA 02138, USA

4 Robert Bosch LLC, Research and Technology Center, Cambridge, MA 02139, USA

Response to the reviewers

Reviewer 1

Reviewer: In the manuscript “Active learning of reactive Bayesian force fields: Application to heterogeneous catalysis dynamics of H/Pt”, the authors demonstrate a numerically efficient approach for approximating an ab initio potential energy surface with a machine learning model for molecular dynamics calculations. The key contribution is i) an end-to-end workflow requiring minimal human interaction and ii) a neat trick to replace a comparison of atomic environments within a polynomial kernel with a fixed tensor product. I suggest publication with minor aspects improved as outlined below.

Authors: We thank the reviewer for their support of the manuscript. We address each of the reviewer’s comments and questions below.

Reviewer: With the promise of machine learning that more data yields more accuracy, it is refreshing to see work on rendering this process affordable and, hence, of practical use. While the key ingredients of the workflow are close to refs 29 and 32 (with some overlap in authors), the computational simplification is a noteworthy and useful result. Making the corresponding code available will certainly help the community to evaluate this approach.

Authors: We thank the reviewer for their support, and note that we have open-sourced the software accompanying this submission to facilitate the evaluation of our method by the community.

Reviewer: The point of the authors that their model is faster than ReaxFF is highly implementation dependent.

Authors: While we agree with the reviewer that our comparison with ReaxFF is *system-dependent*, depending in particular on the number of chemical species in the system (discussed in more detail in the next bullet point), we disagree that the specific cost comparison presented in the manuscript for the H/Pt system is strongly *implementation-dependent*. We have made the cost comparison for the two-element H/Pt system as fair and consistent as possible: both models were evaluated on identical H/Pt structures using the same cpu core and LAMMPS release, and our model is compared against the standard ReaxFF implementation built into LAMMPS. Our goal with this comparison is to challenge the commonly held view that machine learning force fields, while generally more accurate than classical alternatives, are significantly more expensive. The data presented in Fig. S3 establish that, *for the specific reactive H/Pt system studied in this work*, our machine learning force field is considerably faster than the classical force field model of Ref. [1] as implemented in LAMMPS.

To clarify that our comparison is specific to the H/Pt system, we have modified our description of the comparison in the introduction.

Original:

Our prediction speed exceeds that of ReaxFF by more than a factor of two while achieving much higher near-quantum accuracy.

Changed to:

For the H/Pt system, our prediction speed exceeds that of ReaxFF by more than a factor of two while achieving much higher near-quantum accuracy.

Reviewer: Can the authors comment on the formal scaling behaviour of their method with e.g. number of elements, increasing disorder, number of atoms? This would strengthen the argument and would guide the reader.

Authors: We thank the reviewer for these important questions. Regarding the formal scaling with respect to the number of elements, the atomic cluster expansion descriptor vector \mathbf{d} described in Eq. (5) of the main text grows *quadratically* with the number of species N_s , with the precise number of descriptors n_{desc} given by Eq. (7) of the revised manuscript. Because energy evaluations with the $\xi = 2$ mapped model require evaluation of the matrix product $\beta\tilde{\mathbf{d}}$ (see Eq. (22) of the revised manuscript), the prediction cost of the model is expected to grow quadratically with the number elements of the descriptor vector, resulting in an expected $O(N_s^4)$ scaling of the prediction cost with the number of species. We note that if we could construct a descriptor vector \mathbf{d} whose length is *independent* of the number of elements, the prediction time of our models would also be independent of the number of elements. This is a topic that we are actively investigating, but it falls outside the scope of the current work.

We empirically investigate the scaling of our models with respect to the number of descriptor elements and species in Supplementary Figure S15 of the revised manuscript, which reports timings of our mapped models as a function of the number of descriptors for $N_{\text{species}} = 1, 2$, and 3, where N_{species} is the number

of species. We note that quadratic scaling with respect to the number of descriptors is not observed until the number of descriptors exceeds about ~ 500 , which we attribute to the fact that calculating the descriptor and its gradients constitutes a significant fraction of the total prediction time when the number of descriptors is small (as shown in Fig. S16 of the revised manuscript). As a result, the computational cost of the two-species H/Pt model is much less than $2^4 = 16$ -fold the cost of the corresponding one-species model obtained by treating H and Pt as identical species (as shown in Fig. S15 of the revised manuscript).

We have drawn attention to the scaling with respect to the number of species in the Results section:

Original:

The vector \mathbf{d}_i corresponds to the B_2 term in the multielement atomic cluster expansion and is closely related to the SOAP descriptor [2].

Changed to:

The vector \mathbf{d}_i corresponds to the B_2 term in the multielement atomic cluster expansion and is closely related to the SOAP descriptor [2]. In each of these approaches, the number of elements of \mathbf{d}_i scales quadratically with the number of chemical species in the system.

We have also drawn attention to the scaling with respect to the number of descriptors in the Methods section:

Original:

Evaluating the local energy with Eq. (21) requires a single dot product rather than a dot product for each sparse environment, accelerating local energy prediction with the SGP by a factor of N_S . For $\xi = 2$, mean predictions are quadratic in the descriptor and can be evaluated with a vector-matrix-vector product,

$$\varepsilon_{\xi=2}(\rho_i) = \tilde{\mathbf{d}}^\top \boldsymbol{\beta} \tilde{\mathbf{d}}. \quad (1)$$

Changed to:

Evaluating the local energy with Eq. (21) requires a single dot product rather than a dot product for each sparse environment, accelerating local energy prediction with the SGP by a factor of N_S . For $\xi = 2$, mean predictions are quadratic in the descriptor and can be evaluated with a vector-matrix-vector product,

$$\varepsilon_{\xi=2}(\rho_i) = \tilde{\mathbf{d}}^\top \boldsymbol{\beta} \tilde{\mathbf{d}}. \quad (2)$$

The cost of the matrix-vector product $\boldsymbol{\beta} \tilde{\mathbf{d}}$ scales quadratically with the number of descriptors and becomes the principal bottleneck when the number of descriptors is large, as shown in Figs. S15 and S16.

Regarding the scaling of the method with respect to disorder, we first make the general observation that the number of sparse environments needed to obtain a

stable force field during on-the-fly training increases with the diversity of atom-centered chemical environments present in the system. This is established in Table I of the manuscript, where we show that stable dynamics are achieved for the one-element Pt and H₂ systems with fewer than two hundred sparse environments due to considerable redundancy in the chemical environments of the atoms in each frame. It is difficult to predict in advance the precise number of sparse environments that will be needed to obtain a reliable force field, but this is a virtue of active learning methods, which automatically determine if additional environments are needed based on the uncertainty of the current model.

We draw attention to the effect of chemical environment diversity on the number of collected sparse environments in our discussion of our on-the-fly training simulations in Section B of Results:

Original:

The majority of DFT calls were made in the reactive H/Pt(111) simulation, with 216 calls made in total during the 3.7 ps simulation, resulting in nearly 50,000 training labels in total (consisting of the potential energy, atomic forces, and six independent stress tensor components from each DFT calculation).

Changed to:

The majority of DFT calls were made in the reactive H/Pt(111) simulation, with 216 calls made in total during the 3.7 ps simulation, resulting in nearly 50,000 training labels in total (consisting of the potential energy, atomic forces, and six independent stress tensor components from each DFT calculation). The order-of-magnitude increase in the number of accumulated sparse environments during the H/Pt training simulation reflects the greater diversity of chemical environments present in this two-phase simulation, encompassing gaseous H₂, H surface and sub-surface diffusion, and Pt surface and bulk dynamics.

Finally, regarding the scaling of the method with respect to the number of atoms, we note that both during training with our SGP models and during production simulations with our mapped LAMMPS models, the cost of model predictions scales *linearly* with the number of atoms. This is a direct consequence of local energy decomposition, common to most force fields, which expresses the energy of the system as a sum over atom-centered contributions that depend on a small set of neighbors within a local cutoff radius.

We have emphasized this point in our discussion of the energy model in Methods:

Original:

As in the Gaussian Approximation Potential formalism [3], we express the total potential energy $E(\vec{r}_1, \dots, \vec{r}_N; s_1, \dots, s_N)$ of a structure of N atoms as a sum over local energies ε assigned to atom-centered local environments,

$$E(\vec{r}_1, \dots, \vec{r}_N; s_1, \dots, s_N) = \sum_i^N \varepsilon(s_i, \rho_i). \quad (3)$$

Changed to:

As in the Gaussian Approximation Potential formalism [3], we express the total potential energy $E(\vec{r}_1, \dots, \vec{r}_N; s_1, \dots, s_N)$ of a structure of N atoms as a sum over local energies ε assigned to atom-centered local environments,

$$E(\vec{r}_1, \dots, \vec{r}_N; s_1, \dots, s_N) = \sum_i^N \varepsilon(s_i, \rho_i). \quad (4)$$

The decomposition of the total energy into a sum over local, atom-centered contributions gives the model the key property of scaling linearly with the number of atoms in the system.

Reviewer: Considering Table 1, the interface itself seems to be dominating the training cost. How do the authors expect the required training data to behave in the “more complex heterogeneous catalysis” applications the discussion points out, e.g. with adding more surface features or surface size?

Authors: The primary factor determining the amount of data required to train a stable force field with our method is the *geometric and chemical diversity of atom-centered environments*. Based on this principle, we would expect a significant increase in the number of required sparse environments if we were to introduce an additional chemical species into the system, e.g. by modeling a bimetallic alloy instead of a simple metal, or if we were to introduce more complex structural motifs, e.g. nanoparticles. We note, however, that simply increasing surface *size* without introducing geometric or chemical novelty will not increase the required training data, as this duplicates environments already present in the system.

Increased complexity poses a few key challenges to our approach: first, as the reviewer mentions, it increases the time-to-solution of building the model, as more training data must be collected; second, it pushes the memory limits of our Bayesian models, as SGP models require the descriptors of the training environments to be stored in memory; and third, increasing the number of elements significantly slows down our mapped models due to the quadratic scaling of the descriptor with the number of species. There are a number of promising solutions to these problems that we are pursuing, including parallelizing our training protocols to trade time for memory, distributing our SGP models across MPI processes, and investigating more compact descriptors for multi-element systems.

We have made note of some of these promising directions in the Discussion:

Original:

By simplifying the training procedure and reducing the computational cost of reactive force fields at the same time, this unified approach will help extend the range of applicability and predictive power of reactive MD as a modeling

tool, providing new insights into complex systems that have so far remained out of reach.

Changed to:

By simplifying the training procedure and reducing the computational cost of reactive force fields at the same time, this unified approach will help extend the range of applicability and predictive power of reactive MD as a modeling tool, providing new insights into complex systems that have so far remained out of reach. Identifying more compact descriptors for many-element systems and distributing automated training protocols across many processors are promising directions that would help extend the method presented here to systems of even greater complexity. Of particular interest are applications to biochemical reactions and more complex heterogeneous catalysis.

Reviewer: In Figs 3d and 3e the model seems to yield reasonable values for other facets, which is reassuring for somewhat similar local atomic environments, which would explain why the uncertainty for the hollow site for Pt(110) is so high. To me it therefore is counterintuitive that Fig 3b seems to show a substantial error and uncertainty for H₂ distances > 1.3Å that are likely to occur considering the cell in Fig 4a. Can the authors comment on why this did not trigger additional DFT calculations or why these distances are not in the training data?

Authors: To address the reviewer’s question, we have included in the Supplementary Information a plot of the radial distribution function of the H₂ training simulation averaged over all frames (Fig. S3 of the revised manuscript). We observe a sharp peak centered at the equilibrium H₂ bond distance of about 0.74 Å. This peak quantifies the range of H₂ bond lengths observed during the simulation. Crucially, the peak falls to zero at about 1.0 Å and is nearly zero between 1.0 and 1.5 Å (see the inset of Fig. S3). The subsequent rise in the RDF is due to interactions between distinct H₂ molecules. This explains the uncertainty profile observed in Fig. 4(a): the model was not introduced to H₂ bond lengths greater than about 1.0 Å during the on-the-fly training simulation, and therefore the model uncertainty increases for these values of the separation.

We have included a note about this supplementary figure in the Results section:

Original:

The confidence region expectedly grows for extreme bond lengths and dimer separations that were not encountered during training.

Changed to:

The confidence region expectedly grows for extreme bond lengths and dimer separations that were not encountered during training (see Fig. S3 for the radial distribution function of the H₂ training simulation averaged over all frames).

Reviewer 2

Reviewer: In this paper, Vandermause et al. describe the application of reactive Bayesian force-fields to H₂ dissociation and recombination on Pt(111). The paper is well written and quite interesting and thus certainly publishable in principle.

Authors: We thank the reviewer for their support of the manuscript. We address each of the reviewer’s comments and questions below.

Reviewer: I have some comments though: 1) I think that the methodological advances in this paper are not very significant. Unlike the previous FLARE model by the authors, a sparse GP model is used. But (as the authors note) this technology has been used in the GAP method for a long time. In fact, the current method is basically a reimplementaion of GAP with only rather small technical differences. Furthermore, the work of Kresse and co-workers (e.g. refs [33] and [34]) uses a very similar approach in the context of on-the-fly MD simulations. Although these papers are cited, I feel like these close relationships are not adequately acknowledged in the text.

Authors: We agree with the reviewer that our manuscript builds on GAP and the SGP-based on-the-fly method of Kresse and co-workers. A key methodological innovation of this manuscript is the demonstration that these models can be losslessly mapped onto simpler and more efficient polynomial models, which makes them much more attractive in practice.

Additional methodological innovations in our SGP force fields include the use of the atomic cluster expansion of Ralf Drautz for the descriptor, which is cheaper to compute and captures higher body orders than SOAP, the use of a simple normalized uncertainty metric (defined in Eq. (18)) that complements the uncertainty metrics used by Kresse and co-workers, the explicit optimization of SGP hyperparameters by maximizing the log marginal likelihood, including a justification of our choice of kernel and descriptor settings, and the extension of SGP uncertainties to linear combinations of energies, such as surface energies and binding energies (Eq. (15)). We argue that these technical advancements are significant, enabling our recent demonstration of state-of-the-art uncertainty-aware machine learning reactive molecular dynamics of much larger system sizes (0.5 trillion atoms) and higher computational speed (10.5 million atoms-steps/s/node) compared to previous approaches [4].

We have emphasized the relationship of our manuscript to these earlier works in the Introduction of the revised manuscript:

Original:

Our method builds on our own previous active learning workflow, which relied on a significantly more expensive exact Gaussian process [5] and was limited to two- and three-body interactions that are insufficiently descriptive for chemically reactive systems.

Changed to:

Our method builds on the sparse Gaussian process force fields introduced in Refs. [6, 3] and the SGP-based on-the-fly training methods of Refs. [7, 8], extending these methods to a canonical chemically reactive system and establishing their equivalency to a simpler class of polynomial models. Our method also builds on our own previous active learning workflow, which relied on a significantly more expensive exact Gaussian process [5] and was limited to two- and three-body interactions that are insufficiently descriptive for chemically reactive systems.

Reviewer: 2) The main novelty of the approach is the polynomial mapping, which allows accelerating the simulations once the training is completed. In this context, the authors write that “[e]valuating the local energy with Eq. (20) requires a single dot product rather than a dot product for each sparse environment, accelerating local energy prediction with the SGP by a factor of N_S . For $\xi = 2$, mean predictions are quadratic in the descriptor and can be evaluated with a vector-matrix-vector product”. This sounds nice, but the acceleration by a factor N_S is only valid for the linear kernel, which the authors do not use and no statement about the acceleration for the quadratic kernel is made.

Authors: We thank the reviewer for this observation and agree that a more thorough discussion of the $\xi = 2$ case is warranted. We have included in the Methods section of the revised manuscript a precise description of the expected speed up both for $\xi = 2$ and for general ξ :

Added:

Eq. (22) provides an acceleration of SGP local energy prediction if the number of sparse environments exceeds the dimensionality of the descriptor, $N_s > n_{\text{desc}}$, with quadratic prediction expected to be faster by a factor of $\frac{N_s}{n_{\text{desc}}}$. For general ξ , this ratio of efficiencies is equal to $\frac{N_s}{n_{\text{desc}}^\xi}$ and diminishes rapidly with ξ . In our H/Pt models, for which $N_s = 2424$ and $n_{\text{desc}} = 544$, we found $\xi = 2$ models to give considerable improvement over $\xi = 1$, but found no benefit for $\xi \geq 3$ (see Supplementary Table I). We therefore selected $\xi = 2$ for our final trained model and used quadratic prediction when performing production MD simulations, giving a theoretical speedup over SGP mean prediction of $\frac{N_s}{n_{\text{desc}}} \approx 4.5$. In practice we observe a speed up of greater than 10x due to better optimization of the quadratic LAMMPS model (see Fig. S4).

Reviewer: More importantly, this comparison assumes that kernel evaluation rather than construction of the descriptor is the rate-limiting step for the MD simulation. In my experience, computing the descriptor and its derivatives takes a significant part of the computational effort though. The authors should therefore provide more robust data on how much the polynomial form actually accelerates the method.

Authors: We have reported in Fig. S16 of the revised manuscript the fraction of the total prediction time spent computing descriptors and their gradients for our

SGP models and quadratic LAMMPS models. For the quadratic models, we find that the descriptor cost dominates the total cost only for very small descriptor sizes. As the descriptor grows, the fraction of time spent computing descriptors drops to zero, consistent with the quadratic scaling of local energy evaluations with the size of the descriptor (mentioned after Eq. (22) of the revised manuscript and shown empirically in Fig. S15 for large descriptor vectors). For the descriptor size used in this work, $n_{\text{desc}} = 544$, the time spent computing descriptors is under 20%, with the majority of time spent evaluating the matrix product defined in Eq. (22).

Similarly, Fig. S16 shows that for SGP models with sparse set sizes greater than about 800, the majority of time is spent evaluating kernels, not descriptors. As the size of the sparse set increases, the fraction of time spent evaluating descriptors decreases. For $n_{\text{sparse}} = 2010$, which is slightly lower than the number of sparse environments used in our H/Pt model, the descriptor fraction is below 20% for nearly all of the descriptor sizes shown in the figure.

Reviewer: 3) As always with iterative training schemes, I wonder how the authors decided that the training was completed?

Authors: Training can be safely terminated when the force field is able to take many consecutive molecular dynamics steps on its own without making a call to DFT, which implies that it is capable of generating stable dynamics at the target temperature. In practice, we fix the total on-the-fly simulation time in advance and run the simulation to completion. As reported in Table I, we chose values ranging between 3 to 10 ps for the four independent training simulations that we considered. When the training simulation is complete, we confirm by eye that the model was able to take many consecutive steps on its own without making a DFT call (which can be confirmed in Fig. 2(b) and Fig. S2).

We have added a sentence about our termination criteria to the Results section of the revised manuscript:

Added:

The training simulation is terminated when calls to DFT become infrequent, typically after 3-10 ps of dynamics.

References

- [1] Lili Gai, Yun Kyung Shin, Muralikrishna Raju, Adri C. T. van Duin, and Sumathy Raman. Atomistic Adsorption of Oxygen and Hydrogen on Platinum Catalysts by Hybrid Grand Canonical Monte Carlo/Reactive Molecular Dynamics. *JOURNAL OF PHYSICAL CHEMISTRY C*, 120(18):9780–9793, MAY 12 2016.
- [2] Albert P Bartók, Risi Kondor, and Gábor Csányi. On representing chemical environments. *Physical Review B*, 87(18):184115, 2013.

- [3] Albert P Bartók, Mike C Payne, Risi Kondor, and Gábor Csányi. Gaussian approximation potentials: The accuracy of quantum mechanics, without the electrons. *Physical review letters*, 104(13):136403, 2010.
- [4] Anders Johansson, Yu Xie, Cameron J Owen, Jin Soo, Lixin Sun, Jonathan Vandermause, Boris Kozinsky, et al. Micron-scale heterogeneous catalysis with bayesian force fields from first principles and active learning. *arXiv preprint arXiv:2204.12573*, 2022.
- [5] Jonathan Vandermause, Steven B Torrisi, Simon Batzner, Yu Xie, Lixin Sun, Alexie M Kolpak, and Boris Kozinsky. On-the-fly active learning of interpretable bayesian force fields for atomistic rare events. *npj Computational Materials*, 6(1):1–11, 2020.
- [6] Chris M Handley, Glenn I Hawe, Douglas B Kell, and Paul LA Popelier. Optimal construction of a fast and accurate polarisable water potential based on multipole moments trained by machine learning. *Physical Chemistry Chemical Physics*, 11(30):6365–6376, 2009.
- [7] Ryosuke Jinnouchi, Ferenc Karsai, and Georg Kresse. On-the-fly machine learning force field generation: Application to melting points. *Physical Review B*, 100(1):014105, 2019.
- [8] Ryosuke Jinnouchi, Jonathan Lahnsteiner, Ferenc Karsai, Georg Kresse, and Menno Bokdam. Phase transitions of hybrid perovskites simulated by machine-learning force fields trained on the fly with bayesian inference. *Physical review letters*, 122(22):225701, 2019.